# AntiFault: A Fault-Tolerant and Self-Recoverable Floating-Point Format for Deep Neural Networks

## Abstract

Artificial Intelligence (AI) is increasingly deployed in safety-critical applications, where reliability is crucial. However, these AI-based systems are vulnerable to soft errors, where even a single bit flip in a critical model parameter can lead to complete system failure. Existing fault-tolerant solutions typically rely on hardware redundancy or additional memory overhead, which is impractical for resource-constrained edge devices. In order to address this, we introduce AntiFault, a novel 16-bit floating-point representation technique that approximates 32-bit floating-point numbers while embedding fault protection within the same bit width without incurring any additional memory overhead. AntiFault ensures minimal to no accuracy degradation and also enables multiple error detection, localization, and correction without requiring additional memory space. Our approach reduces model size by upto 50%, while guaranteeing protection and recovery from soft errors. We have evaluated AntiFault on image and text classification tasks using ResNet18, MobileNetV2, and MobileViT on CIFAR-100 and MNIST, and DistilBERT and RoBERTa on Emotion and AG's News datasets. Experimental results show that models using AntiFault maintain their accuracy under extensive fault injection, while standard 32-bit models suffer severe degradation from even a single-bit flip in critical bits such as sign and/or exponent bits.

## 1 Introduction

In recent years, Artificial Intelligence (AI) has been increasingly deployed in critical applications, where precise and reliable decision-making is essential. Despite extensive training and validation, AI models remain vulnerable to soft errors (transient faults caused by cosmic radiation or voltage fluctuations), that can affect even high-end hardware. Although soft errors do not result in permanent physical damage, they can corrupt critical data during execution, producing incorrect outputs. During inference, model parameters are usually stored in main memory, particularly RAM, which is highly susceptible to such faults Sridharan et al. (2015). A single bit flip in memory can propagate through computations, ultimately leading to incorrect predictions. Thus, soft errors represent a significant threat to the reliability and robustness of AI systems.

To ensure the reliability of AI systems, it is essential to first understand how soft errors manifest and affect the overall system output. Prior studies Goldstein et al. (2020); Bosio et al. (2019); Li et al. (2017); Xue et al. (2023b) have analyzed the behavior of various deep learning models under different soft error rates. These analyses reveal that a model's resilience can vary significantly depending upon the data types, network layers, activation functions, and the specific locations where faults occur. Figure 1 illustrates our problem motivation, demonstrating how soft errors impact AI models across both computer vision (CV) and natural language processing (NLP) tasks under varying bit error rates (BER). It can be observed that, as few as one to three bit flips can cause a substantial drop in model accuracy, highlighting the high vulnerability of these AI models towards soft errors.

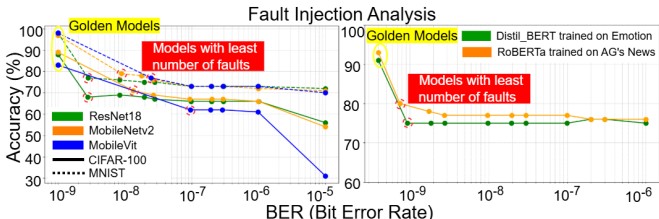

Figure 1: Impact of fault injection on classification accuracy across different deep learning models.

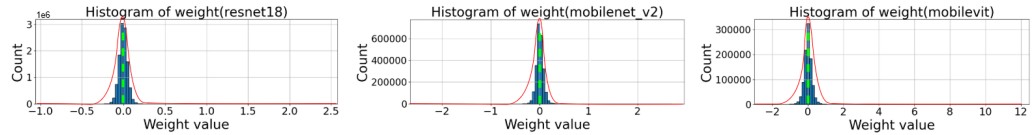

Figure 2: Weight values of Resnet18, MobileNetv2 and MobileVit finetuned on CIFAR100

From past work, it can be seen that the type of datatype been used significantly impacts the robustness of AI models against soft errors. Goldstein et al. (2020) shows that floating-point numbers are more sensitive to soft errors because the number explodes if the exponent bit is hit. Therefore, we need a mechanism to make 32-bit floating-point numbers fault-tolerant without compromising accuracy. Several techniques have been proposed to overcome this problem, such as, Algorithm-Based Fault Tolerance (ABFT) Huang & Abraham (1984); Zhao et al. (2021) and hardware redundancy methods such as Dual/Triple Modular Redundancy (DMR/TMR) Li et al. (2019); Hosseinkhani & Ghavami (2021), and neuron duplication. However, these approaches typically incur significant memory and computational overhead, making them unsuitable for resource-constrained environments such as edge devices. Fault-aware model retraining has also been explored as a mitigation strategy, but it is often impractical due to its intensive computational demands, long training times, and the need for access to original training data. As a result, there is a pressing need for a lightweight, post-training fault-tolerance solution that introduces no additional memory overhead.

In order to address the above-mentioned challenges, we introduce AntiFault, a novel fault-tolerant approximation technique that compresses a 32-bit floating-point number into a 16-bit representation while preserving computational integrity under soft errors. *To the best of our knowledge, this is the first approach that employs an approximation technique to enable full fault tolerance in floating-point representation.*. AntiFault works by identifying the most essential and non-redundant bits from the original 32-bit float, along with using parity bits within the 16-bit format. These parity bits enable real-time detection and correction of faults at any bit position without the need to reload parameters from memory. Unlike traditional fault-tolerance methods that increase memory overhead, AntiFault achieves full fault tolerance with a 50% reduction in model size. Moreover, AntiFault is a post-training solution, enabling models to be made fault-tolerant without requiring retraining. Experimental results show that AntiFault achieves 100% resilience to single-bit errors and also protects and detects double-bit flips in the most critical part, with minimal to no accuracy loss due to approximation.

In particular, the following are the key novel contributions of our paper:

- We conducted parameter value analysis and average bit value analysis across multiple deep learning models to identify commonly used value ranges and determine the most significant bits.

- We proposed an approximation technique that compresses model parameters by 50%, while also introducing parity bits, and preserving the classification accuracy of AI models.

- We developed a dual recovery mechanism that protects all bits in the 16-bit datatype and enables full recovery from single-bit errors and partial recovery in case of double-bit errors.

- We performed extensive fault injection analysis across multiple components of the floating-point format (sign, exponent, and mantissa) for both image and text classification tasks.

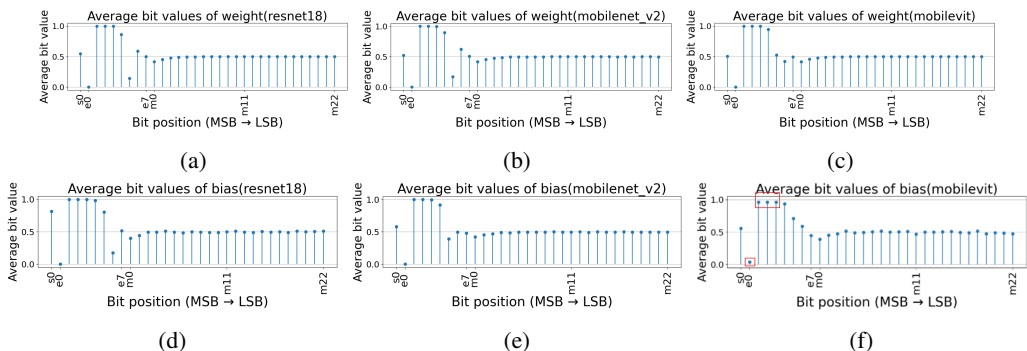

Figure 3: Average bit values of weights and biases of Resnet18, MobileNetv2 and MobileVit pre-trained on CIFAR-100

- We compared the performance of all models using standard 32-bit floating-point representations against our proposed AntiFault format.

## 2 RELATED WORK

Dual Modular Redundancy (DMR) and Triple Modular Redundancy (TMR) are classic hardware-level fault-tolerant techniques that are adopted to improve AI-based system reliability. Li et al. (2019) proposed selectively duplicating the critical layers of a neural network. By comparing the outputs of these duplicated layers, soft errors can be effectively detected with reduced computational and memory costs. Similarly, Hosseinkhani & Ghavami (2021) proposes an approximated triple modular redundancy method to provide varying levels of protection across different layers. However, the effectiveness of these methods depends on the accurate identification of the most vulnerable layers. So far, redundancy-based approaches such as TMR and DMR pose additional hardware overhead in terms of both memory and computation. This limitation make redundancy-based techniques less suitable for resource-constrained or high-efficiency AI systems.

Algorithm-based fault tolerance (ABFT) is a software-level technique proposed by Huang & Abraham (1984) for detecting and correcting soft errors. It utilizes additional checksum rows and columns for detecting and localizing errors. This concept is further extended to non-linear and high-dimensional computations of convolutional neural networks in Zhao et al. (2021), which incorporates a lightweight checksum scheme into convolutional operations to enable efficient fault correction with minimal overhead. Xue et al. (2023a) further optimized ABFT for vision transformers by applying selective error correction by ignoring low-impact faults. While ABFT is generally more efficient than redundancy-based techniques, it requires additional memory to store checksums for all parameters and offers no protection for the checksum data itself, hence remains prone to soft errors.

Wang et al. (2021) proposed FTApprox, a 16-bit fault-tolerant data format that leverages approximate computing for integers and fixed-point numbers. FTApprox cannot locate or correct the fault in data bits, it only detects if a fault has occurred. This work is further extended by Mishra et al. (2023), by making fixed-point approximation more configurable and efficient. It also lacks the ability to correct soft errors in data bits. These methods can not perform well with floating-point numbers due to their complex format; a single bit flip in sensitive parts of a floating-point number, such as the exponent or mantissa, can lead to severe corruption and significant accuracy loss. Moreover, while these formats can detect faults in data bits, they cannot localize or correct them, limiting their effectiveness in fault-sensitive applications.

## 3 OUR NOVEL ANTIFAULT METHODOLOGY

To generate AntiFault, important data bits are identified from a 32-bit floating-point number for approximation. For the protection of data bits, parity bits are calculated that help in the localization and correction of soft errors in data bits. Furthermore, a dual fault detection and recovery mechanism

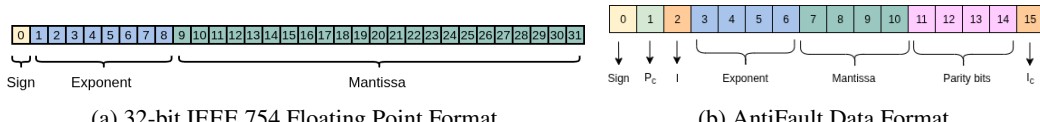

(a) 32-bit IEEE 754 Floating Point Format      (b) AntiFault Data Format

Figure 4: Binary representations of 32-bit IEEE 754 floating point number and AntiFault

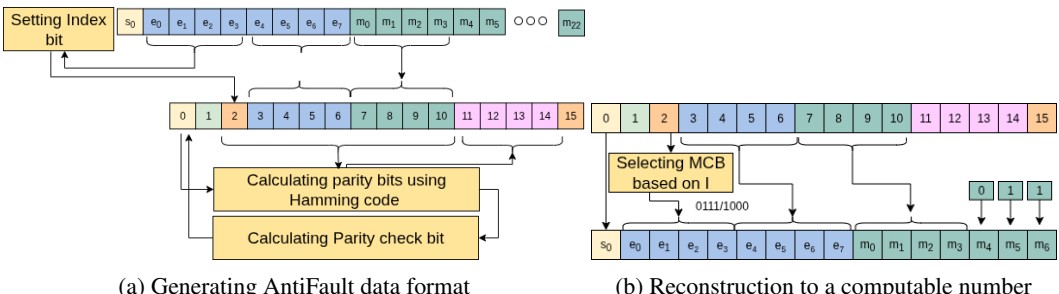

(a) Generating AntiFault data format      (b) Reconstruction to a computable number

Figure 5: Workflow of converting a 32-bit float to Antifault and constructing a computable number from AntiFault

is proposed that can protect and correct all 16 bits of AntiFault. Then, an executable number is constructed from the data bits, and operations are performed, ensuring minimal to no accuracy degradation. Detailed methodology is explained in the following subsections.

## 3.1 IDENTIFYING DATA BITS FOR ANTIFAULT

As shown in the Figure 4a, 32-bit float has 3 components ie. sign, exponent, and mantissa. Let $S = \{s_0\}, E = \{e_0, e_1, e_2, e_3, e_4, e_5, e_6, e_7\}, M = \{m_0, m_1, m_2, m_3, m_4, ....., m_{22}\}$ be the set of sign, exponent and mantissa bits respectively, indexing is done from left to right. For approximating a 32-bit float in a 16-bit data format, we analyzed the values of different parameters in DNNs to identify the bits that are constant throughout all the parameters $\theta$. Parameters of DNNs follow a normal distribution, which can be seen in Figure 2. Weight values across all networks follow a normal distribution having sharp peaks near zero with thin, long tails, indicating that most values are concentrated around zero, but a small number extend farther out.

If the parameters follow a particular distribution, then the bits of these parameters must also follow a certain pattern. Therefore, we plotted the average bit values for weight and bias of all the models. We denote the first four bits of the exponent of a parameter $\theta_i$ as $MCB(\theta_i)$, which represents the *most critical block*. It can be seen in Figure 3, that most parameters have $MCB(\theta_i) = [0, 1, 1, 1]$, but in Figure 3f, it can be seen that there is a portion of values that have different values for $MCB(\theta_i)$. Futhermore, it has been observed that $MCB(\theta_i)$ for all the parameters is either $0, 1, 1, 1$ or $1, 0, 0, 0$. Therefore, we divided the parameters in two ranges depending on their binary representation.

As we know that the $MCB(\theta_i)$ can have only two sets of bits, we exempt it from our proposed data type, as we can add it later in the reconstruction part. Whereas, Sign bits and mantissa bits have a 50% chance of being 0 or 1. So the sign bit and the first 4 mantissa bits from the 32-bit float. $s_0, e_4, e_5, e_6, e_7, m_0, m_1, m_2,$ and $m_3$ are a part of our proposed datatype and act as data bits.

## 3.2 GENERATING 16-BIT ANTIFLOAT FROM 32-BIT FLOAT

Conversion of parameters of a model from floating-point numbers to AntiFault doesn't require any additional training. Once a model is fully trained and ready to deploy, its parameters can be converted to AntiFloat. Let the trained 32-bit AI model be $f_\theta$ with $\theta$ as its parameters and AntiFault model be $f'_{\theta'}$ with $\theta'$ as protected parameters. To suppress numerical noise in $f_\theta$, $\forall \theta_i$ such that $0 < |\theta_i| \rightarrow 0$, $\theta_i$ were explicitly set to $0$. Main components of AntiFault is data bits and parity bits. Data bits consist of a sign bit, exponent bits, mantissa bits, and an indicator bit. While identifying the important data bits from a 32-bit floating point number, we discarded

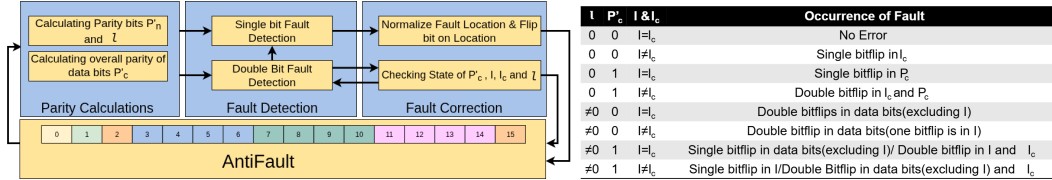

(a) Two-way fault detection and correction mechanism      (b) All possible fault scenarios in AntiFault

Figure 6: Workflow of fault detection and correction mechanism of AntiFault

$MCB(\theta_i)$, so the indicator bit $I$ helps in identifying these bits during the reconstruction phase. $I$ is calculated by equation (1).

$$I = \begin{cases} 0 & \text{if } \theta_i > -2 \text{ and } \theta_i < 2 \\ 1 & \text{if } \theta_i \geq 2 \text{ or } \theta_i \leq -2 \end{cases} \tag{1}$$

The primary goal of this methodology is to protect the data bits, to make the model parameters resilient to soft errors. For this purpose, we have used hamming code Hamming (1950) to calculate parity bits. These parity bits are capable of both detecting faults and identifying the exact bit location of the error, which then can be corrected accordingly. To use this method we need four parity bits to protect an 10-bit sequence, which is determined by equation (2), where $n_p$ is the number of parity bits and $n_d$ is the number of data bits. Let parity bits be $p_0, p_1, p_2$, and $p_3$ and bit sets be $s_0 = [0, 2, 4, 5, 7, 9]$, $s_1 = [0, 3, 4, 6, 7, 10]$, $s_2 = [2, 3, 4, 8, 9, 10]$, $s_3 = [5, 6, 7, 8, 9, 10]$. $p_0, p_1, p_2$, and $p_3$ can be calculated using equation (3), where $\theta_i^{'(j)}$ denotes the $j$-th bit of $i$-th parameter of $f_\theta'$.

$$2^{n_p} \geq n_d + n_p + 1 \tag{2}$$

$$p_n(\theta_i') = \bigoplus_{j \in s_n} \theta_i^{'(j)}, \quad n \in \{0, 1, 2, 3\} \tag{3}$$

These parity bits are responsible for the detection and correction of single bit flips. To enable our data type to detect and correct double bit-flips, we used $P_c$, which will calculate the overall parity of the databits and parity bits. $P_c$ can be calculated using equation (4.

$$P_c(\theta_i') = \bigoplus_{j=0}^{14} \theta_i^{'(j)}, \quad j \neq 1 \tag{4}$$

In $\theta_i'$, the most critical bit is the $I$ as it stores the information of $MCB(\theta_i)$. Therefore, we saved a redundant copy of $I$ as $I_c$. $I_c$ is not considered a data bit and was excluded during the calculation of the parity bits. In Figure 4b, it can be seen that the AntiFault has a sign bit $s_0$, a parity-check bit $P_c$, indicator bit $I$, four exponent bits ($e_4, e_5, e_6, e_7$), four mantissa bits ($m_0, m_1, m_2, m_3$), parity bits($p_0, p_1, p_2, p_3$) and $I_c$. These bits are arranged in such a way that they prevent overflow in the exponent part. The workflow for the generation of AntiFault data type can be seen in Figure 5a

### 3.3 DUAL FAULT DETECTION AND CORRECTION

Every bit in AntiFault is protected by other bits and can also be recovered. For the recovery of the data bits and parity bits, we calculated $p_0', p_1', p_2', p_3'$ using equation (5). Let $l$ be an unnormalized location of fault calculated by equation(7), where $p_0', p_1', p_2'$ and $p_3'$ are concatenated and the result is interpreted as an integer. Values of $l$ and $P_c'$ are used to identify whether there is a single or double bit flip. If $l \neq 0 \& P_c' \neq 0$, it indicates there is a single bit flip in data or parity bits. For data bits, the fault location is normalized using equation (8). For a fault in parity bits, if $l = 2^n$, then the fault is in $p_n$. If $l \neq 0 \& P_c' = 0$, it indicates there is a double bit flip. We considered all scenarios of double bit flips that can affect $I$, given in figure 6b. In all possible cases, we can identify whether there is a fault in Pc, $I$, or $I_c$. In case of a double-bit fault affecting $I$, $I$ can fully be recovered using $I_c$. If we have identified a double-bit fault in $I$, then the other bit location can be identified using $l$, as there is a unique $l$ for all combinations of faults if one fault position is known.

$$p_n'(\theta_i') = \bigoplus_{j \in s_n} \theta_i^{'(j)} \oplus p_n, \quad n \in \{0, 1, 2, 3\} \tag{5}$$

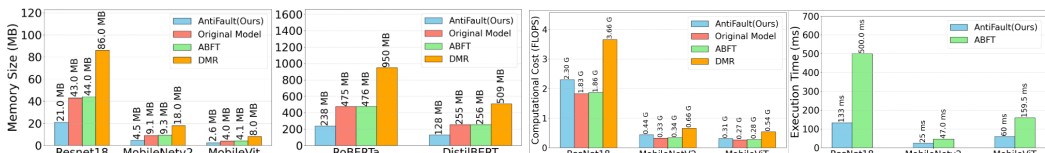

(a) Memory size compression of different fault protection algorithms

(b) Computational overhead compression of different fault protection algorithms

Figure 7: Comparison of memory size and computational overhead of AntiFault with existing fault protection methods

| | No Additional Memory Overhead | Detection of Single Bit Flips | Correction of Single Bit Flips | Detection of 2bit flips | Correction of 2bit flips | Detection of Errors in data used for protection | Correction of Errors in data used for protection |
|---|---|---|---|---|---|---|---|
| DMR | ✗ | ✓ | ✗ | ✗ | ✗ | Redundant model is also vulnerable | ✗ |
| TMR | ✗ | ✓ | ✓ | ✗ | ✗ | Redundant models are also vulnerable | ✗ |
| ABFT (Zhao et al. 2021) | ✗ | ✓ | ✓ | ✓ | ✗ | Checksum bits are also vulnerable | ✗ |
| FTApprox (Wang et al. 2021) | ✓ | ✓ | Only for the location of MSV block | Only for the location of MSV block | ✗ | ✗ | ✗ |
| AntiFault (Ours) | ✓ | ✓ | ✓ | ✓ | Only in most critical bits | ✓ | ✓ |

Figure 8: Qualitative comparison of AntiFault with existing methods

$$P'_c(\theta'_i) = \bigoplus_{j=0}^{14} \theta_i^{'(j)}, \tag{6}$$

$$l = int(concat[p3', p2', p1', p0']) \tag{7}$$

$$\text{faultLocation} = \begin{cases} l - 4 & \text{if } l > 8 \\ l - 3 & l \leq 8 \end{cases} \tag{8}$$

In order to correct the error in data bits, the bit at which the fault has occurred is flipped back to its original value. In AntiFault, data bits are protected and recovered by parity bits, and parity bits, $p_c$ and $I_c$ can also be recovered. Hence, providing a dual protection mechanism, shown in Figure 6a.

After fault detection and correction, the next step is to generate a computable number from data bits. Therefore, we reconstructed a 16-bit float with 1 sign bit, 8 exponent bits, and 7 mantissa bits. Sign bit in AntiFloat is directly mapped to the sign bit in the reconstructed number. AntiFault has only 4 exponent bits $e_4, e_5, e_6, e_7$, whereas we need 8 bits of exponent for reconstruction. Therefore, the indicator bit $I$ is checked to determine $MCB(\theta_i)$, if $I$ is 0, then $[0, 1, 1, 1]$ is assigned to $MCB(\theta_i)$ otherwise, $[1, 0, 0, 0]$ is assigned to $MCB(\theta_i)$. Similarly, AntiFault has only 4 mantissa bits $m_0, m_1, m_2, m_3$, so $0, 1, 1$ are assigned to $m_4, m_5, m_6$. $0, 1, 1$ is added because the experiments, in Appendix A.2, have showed it produced the least error compared to other combinations. Figure 5b shows the reconstruction of a computable number. This 16-bit computable number follows the same method of performing the operations of multiplication and addition, similar to other floating point numbers i.e., float16, float32, and float64.

## 4 EXPERIMENTS AND RESULTS

Firstly, we evaluated the approximation model to assess its compatibility with AI applications. Afterwards, AntiFault and the 32-bit floating-point format were evaluated on both image classification and text classification tasks.

### 4.1 FAULT-INJECTION MODEL

To analyze the behavior of AI models under soft errors, we performed software-level fault injection experiments using single-bit and double-bit flips fault model for both 32-bit float and AntiFault data format. Models using 32-bit floating point numbers without any soft errors are considered

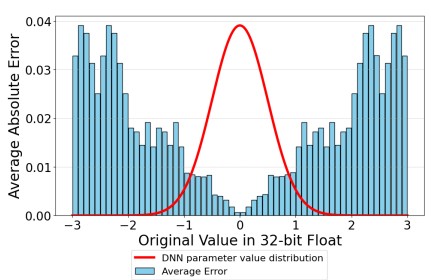

Figure 9: Approximation error on range $[-3, +3]$ with step size of $0.0001$

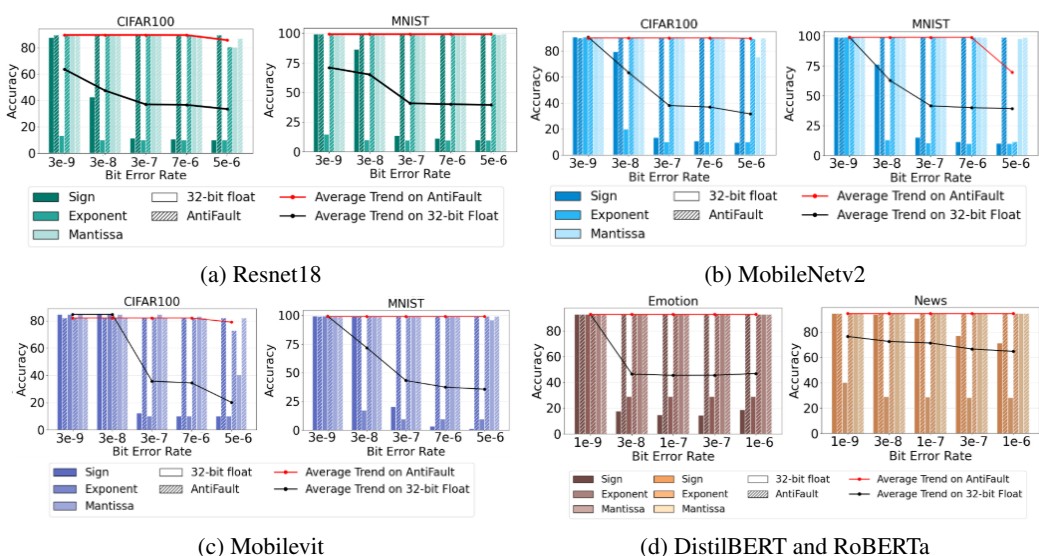

(a) Resnet18

(b) MobileNetv2

(c) Mobilevit

(d) DistilBERT and RoBERTa

Figure 10: Analyzing accuracy degradation on Resnet18, Mobilenet v2 and Mobilevit when faults are injected on different parts (sign, exponent, mantissa) of the data type using 32-bit Float and AntiFaults.

as the golden models and are used as a reference for comparison. Experiments were conducted by injecting faults during the inference phase. We have evaluated the models through varying Bit Error Rate (BER) from $10^{-9}$ to $10^{-6}$ on different parts of the data format i.e., sign, exponent and mantissa bits, to assess the sensitivity of each component. For specific BER, new random bit-flips were injected before the execution of each test batch for a more dynamic and random fault pattern throughout the evaluation process in a more efficient way. BER is calculated by equation 9, where $n_f$ is the total number of faults in the model and $N$ is the total number of bits in the model.

$$BitErrorRate(BER) = \frac{n_f}{N} \tag{9}$$

### 4.2 APPROXIMATION MODEL

In AntiFault data format, the 32-bit floating point number is approximated to 16-bit AntiFault. To evaluate the error introduced by this approximation, we conducted a controlled simulation over the range of [-3,+3], which covers most of the dynamic range of the parameters in DNNs. This range is sampled with a step size of 0.0001 to create a high resolution set of input values. For the experimentation, each value is converted to 16-bit AntiFault and then reconstructed back to a 16-bit computable floating point number. The approximation error is calculated as the average absolute difference between the original values and reconstructed values across the entire range. The results of this experimentation are plotted in Figure 9. As we can see, the values closer to zero have the minimum error, and as the value moves farther, the larger the approximation error gets. So we can say that this approximation model will guarantee performance for AI models, as in Figure 2, most

of the parameters of AI models are centered on zero. Hence, models whose values follow a sharp peak normal distribution centered on zero will have the least approximation error.

## 4.3 IMAGE CLASSIFICATION

To evaluate AntiFault on image classification, we conducted experiments on convolutional neural networks (CNNs) and vision transformer using CIFAR100 Krizhevsky et al. (2009) and MNIST LeCun et al. (1998) datasets. For CNN architectures, we selected ResNet18 He et al. (2016) and MobileNetV2 Sandler et al. (2018), and for vision transformer, we have used MobileVit Mehta & Rastegari (2021).

Results of fault injection on the classification model using 32-bit floating point numbers and AntiFault can be seen in Table 1. In Figure 10, it can be seen that the most critical part of a data type is its exponent part. A single bit flip in the exponent part can significantly degrade the accuracy of the model. A single sign bit also holds great significance; it can be seen that all the models experience a significant dip in the accuracy if faults are injected only on the sign bit at a BER of $10^{-6}$. Unlike the exponent and the sign bit, faults at the mantissa has the least effect on the accuracy of the model.

The golden model accuracies for ResNet18, MobileNetV2, and MobileVit are 89.81%, 90.56% and 84.65%, respectively, on the CIFAR-100 dataset. While using AntiFault ResNet18, MobileNetV2, and MobileVit had an accuracy loss of 0.2%, 0.66%, and 2.64% respectively, due to approximation error. However it can be seen in Table 1 and in Figure 10, the accuracy of ResNet18 and MobileVit while using AntiFault remains constant across BER from $10^{-9}$ to $10^{-6}$ due to its robustness to all single-bit flips and double bitflips in $MCB_\theta$. When AntiFault is tested on these models using the MNIST dataset, very little accuracy is dropped due to approximation. ResNet18, MobileNetV2 and MobileVit had an accuracy drop of 0.07%, 0.16% and 0.14%, respectively. In these experiments, AntiFault maintained the accuracy of models across the whole BER range except for MobileNetv2, which dropped the accuracy at $5 \times 10^{-6}$ due to heavy double-bit fault injection in the exponent

## 4.4 TEXT CLASSIFICATION

For text classification, AntiFault is evaluated on two state-of-the-art transformer models: DistilBERT Sanh et al. (2019) and RoBERTa Liu et al. (2019). DistilBERT is tested on the Hugging Face Emotion dataset Zhu et al. (2018), and for RoBERTa, evaluation is performed on the AG News dataset Zhang et al. (2015).

DistilBERT and RoBERTa are evaluated using both standard 32-bit floating point representation and AntiFault. The results are summarized in Table 1 and visualized in Figure 9d. DistilBERT shows more robustness to soft errors occurring in the sign bit compared to other models. However, it crashes significantly when a soft error strikes the exponent part of the floating-point representation. Whereas, with the AntiFault, DistilBERT maintains its accuracy across a wide range of Bit Error Rates (BER), from $10^{-9}$ to $10^{-6}$, with no degradation in accuracy due to approximation.

From the results, it can also be observed that RoBERTa begins to show a drop in accuracy at a BER of $10^{-9}$ when faults affect the exponent part. At the same BER, the sign and mantissa bits remain relatively robust. However, as the BER increases, the accuracy degrades significantly when faults strike the sign bit. However, the mantissa remains resilient even up to a BER of $10^{-6}$, showing a lower sensitivity to bit flips in that region. The accuracy loss due to the approximation introduced by the AntiFault is 0.06, making it almost negligible. AntiFault enhances DitilBERT's and RoBERTa's robustness against soft errors without affecting model performance.

## 4.5 MEMORY AND COMPUTATIONAL OVERHEAD

Unlike existing fault-tolerant techniques that rely on redundant copies or checksum storage, AntiFault uses 50% less memory by embedding protection bits within the same bitwidth, shown in Figure 7a. However it has a computation overhead due to fault detection and correction of all bits. It has higher computation overhead than ABFT but lower than DMR and TMR. We implemented AntiFault for both CPU and GPU. But on GPUs, it outperforms ABFT in execution time, as can be seen in Figure 7b. Selective use of AntiFault is described in Appendix A.3.

Table 1: Comparing results of fault injection on image and text classification models using 32-bit float and AntiFault

| Network (Dataset) | Location | Method | $3 \times 10^{-9}$ | $3 \times 10^{-8}$ | BER $3 \times 10^{-7}$ | $7 \times 10^{-7}$ | $5 \times 10^{-6}$ |
|---|---|---|---|---|---|---|---|
| ResNet18 (CIFAR-100) | Sign | 32-bit Float | 87.7 | 42.51 | 11.29 | 10.6 | 10 |
| | | AntiFault | **89.61** | **89.61** | **89.61** | **89.61** | **89.61** |
| | Exponent | 32-bit Float | 13.29 | 10.01 | 10.0 | 10.0 | 10.0 |
| | | AntiFault | **89.61** | **89.61** | **89.61** | **89.41** | **80.42** |
| | Mantissa | 32-bit Float | **89.79** | **89.75** | 89.57 | 89.14 | 80.21 |
| | | AntiFault | 89.61 | 89.61 | **89.61** | **89.5** | **87.04** |
| MobileNetV2 (CIFAR-100) | Sign | 32-bit Float | **90.57** | 79.37 | 13.43 | 10.71 | 9.54 |
| | | AntiFault | 89.9 | **89.9** | **89.9** | **89.9** | **89.89** |
| | Exponent | 32-bit Float | **90.57** | 19.96 | 10.0 | 9.98 | 9.97 |
| | | AntiFault | 89.9 | **89.9** | **89.9** | **89.9** | **89.28** |
| | Mantissa | 32-bit Float | **90.57** | **90.59** | **90.52** | 89.85 | 75.52 |
| | | AntiFault | 89.9 | 89.9 | 89.9 | **89.9** | **89.84** |
| MobileVit (CIFAR-100) | Sign | 32-bit Float | **84.65** | **84.65** | 12.34 | 10.09 | 10.12 |
| | | AntiFault | 81.97 | 81.97 | **81.97** | **81.97** | **81.97** |
| | Exponent | 32-bit Float | **84.65** | **84.65** | 9.97 | 10.0 | 10.0 |
| | | AntiFault | 81.97 | 81.97 | **81.97** | **81.97** | **72.82** |
| | Mantissa | 32-bit Float | **84.65** | **84.65** | **84.58** | **83.87** | 63.26 |
| | | AntiFault | 81.97 | 81.97 | 81.97 | 81.97 | **81.19** |
| ResNet18 (MNIST) | Sign | 32-bit Float | 99.32 | 86.32 | 13.52 | 11.31 | 9.98 |
| | | AntiFault | **99.25** | **99.25** | **99.25** | **99.25** | **99.25** |
| | Exponent | 32-bit Float | 14.58 | 9.96 | 9.8 | 9.8 | 9.8 |
| | | AntiFault | **99.25** | **99.25** | **99.25** | **99.3** | **99.16** |
| | Mantissa | 32-bit Float | 99.22 | 99.22 | 99.22 | 99.22 | 98.79 |
| | | AntiFault | **99.25** | **99.25** | **99.25** | **99.25** | **99.21** |
| MobileNetV2 (MNIST) | Sign | 32-bit Float | **98.89** | 76.17 | 15.14 | 11.44 | 10.26 |
| | | AntiFault | 98.73 | **98.73** | **98.73** | **98.73** | **98.73** |
| | Exponent | 32-bit Float | **98.89** | 11.81 | 9.86 | 9.8 | 9.8 |
| | | AntiFault | 98.73 | **98.73** | **98.73** | **98.73** | **11.35** |
| | Mantissa | 32-bit Float | **98.89** | **98.87** | 98.9 | **98.78** | 97.63 |
| | | AntiFault | 98.73 | 98.73 | **98.73** | 98.73 | **98.72** |
| MobileVit (MNIST) | Sign | 32-bit Float | **99.35** | 98.79 | 10.48 | 2.32 | 1.02 |
| | | AntiFault | 99.21 | **99.21** | **99.21** | **99.21** | **99.21** |
| | Exponent | 32-bit Float | **99.35** | 11.33 | 9.85 | 9.8 | 9.8 |
| | | AntiFault | 99.21 | **99.21** | **99.21** | **90.1** | **99.1** |
| | Mantissa | 32-bit Float | **99.35** | **99.35** | **99.33** | 99.1 | 98 |
| | | AntiFault | 99.21 | 99.21 | 99.21 | **99.21** | **99.21** |

| | | | $10^{-9}$ | $3 \times 10^{-8}$ | BER $10^{-7}$ | $3 \times 10^{-7}$ | $10^{-6}$ |
|---|---|---|---|---|---|---|---|
| DistilBERT (Emotion) | Sign | 32-bit Float | 92.7 | 17.8 | 14.7 | 14.5 | 18.6 |
| | | AntiFault | **92.7** | **92.7** | **92.7** | **92.7** | **92.7** |
| | Exponent | 32-bit Float | 92.7 | 29.05 | 29.05 | 29.05 | 29.05 |
| | | AntiFault | **92.7** | **92.7** | **92.7** | **92.7** | **92.7** |
| | Mantissa | 32-bit Float | **92.7** | 92.45 | 92.55 | **93.05** | **92.85** |
| | | AntiFault | **92.7** | **92.7** | **92.7** | 92.7 | 92.7 |
| RoBERTa (AG's News) | Sign | 32-bit Float | **94.6** | 94.03 | 90.9 | 77.02 | 71.45 |
| | | AntiFault | **94.6** | **94.6** | **94.6** | **94.6** | **94.6** |
| | Exponent | 32-bit Float | 40.4 | 29.06 | 28.7 | 28.3 | 28.16 |
| | | AntiFault | **94.6** | **94.6** | **94.6** | **94.6** | **94.6** |
| | Mantissa | 32-bit Float | **94.69** | **94.69** | **94.69** | **94.67** | **94.65** |
| | | AntiFault | 94.6 | 94.6 | 94.6 | 94.6 | 94.6 |

## 5 CONCLUSION

In this paper, we propose AntiFault, a 16-bit data format, for AI models that provides full protection against single-bit flips and partial protection against double-bit flips. AntiFault embeds parity bits within the same 16-bit space, without needing additional memory. AntiFault uses a dual protection mechanism where each bit is protected by other bits, enabling complete recovery from any single-bit fault. We evaluated the robustness of AntiFault against 32-bit float under varying bit error rates. Experimental results show that 32-bit float suffers significant accuracy degradation when faults occur in critical bits (e.g., sign or exponent), whereas AntiFault maintains model accuracy even under extensive fault injection. AntiFault demonstrates that a system can be fully protected from soft errors without requiring additional memory by carefully approximating large data formats into a smaller format and utilizing the freed-up space to protect the important bits. While the current design has some computational overhead, we plan to optimize our approach to further minimize it.

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
