# A  APPENDIX

This Section consists of the implementation details for the image classification and text classification models used in the paper. It also contains the experimental details of how we identified the best-performing combinations for the last three bits of the mantissa for the computable number. In the final section, we discuss the flexibility of AntiFault and how it can be applied selectively to specific parts of an AI model to achieve minimal computational overhead.

## A.1  IMPLEMENTATION DETAILS

### A.1.1  RANDOM SEED

To ensure random fault injections, we performed dual seeding. First seed of 42 is given to generate a random number. Then a random number is generated in the range [1,100] and is used as the final seed to generate random bit flips before inference of each batch.

### A.1.2  IMAGE CLASSIFICATION

All models were initialized with pretrained ImageNet weights, and the final classification layer was replaced to match the number of classes in the target datasets. We conducted experiments on two datasets ie. CIFAR-100 and MNIST. Each model is fine-tuned for 5 epochs using the Adam optimizer and a batch size of 64, with cross-entropy loss as the objective function. We implemented our experiments using PyTorch and utilized the timm library for model loading and fine-tuning.

Both the datasets CIFAR-100 consists of 60,00 images categorized into 100 classes with 10,000 test images. MNIST consist of 70,000 gray-scale images, of handwritten digits, classified into 10 classes.are resized to 224x224 and normalized. MNIST is converted to RGB images by duplicated channels.

### A.1.3  TEXT CLASSIFICATION

Pretrained models of textattack/roberta-base-ag-news and bhadresh-savani/distilbert-base-uncased-emotion are used which are finetuned for their respective classification tasks. For preprocessing, tasks like lower casing, token truncation, padding and addition of special tokens are handled by Hugging Face tokenizers to ensure that the input representation is consistent with the representation the models are originally trained on.

For experiments, the test split of the Hugging Face Emotion dataset is used. This dataset consists of 2,000 text samples categorized into six emotion classes: sadness, joy, love, anger, fear, and surprise. The test split of the AG News dataset, which contains 7,600 news samples distributed across four categories: World, Sports, Business, and Science/Technology.

## A.2  EXPERIMENTATION FOR IDENTIFYING THE LAST 3 BITS OF THE COMPUTABLE NUMBER

In the AntiFault format, we only use 4 bits for the mantissa, but the computable number requires 7 mantissa bits. We conducted controlled simulations to determine the best values for last three missing mantissa bits.

We generated a set of input values ranging from -1 to +1, with step size of 0.0001. This gave us a large number of test points to analyze. For each value, we created a 16-bit computable floating-point number by appending all possible 8 3-bit combinations to the existing 4-bit mantissa. This allowed us to test which 3-bit extensions resulted in the most accurate approximations.

To measure the approximation error, we used the average absolute error which is, the average of the absolute differences between the original values and the reconstructed values over the entire range. Results of this experiment can be seen in in Table 1. The 3-bit patterns 011, 100, and 101 resulted in the lowest errors. Therefore, any one of these bit patterns can be used as the last three bits of the mantissa to minimize error during approximation.

Table 1: Error while using different combinations for last three bits of mantissa

| 3rd last bit | 2nd last bit | last bit | Error |
|---|---|---|---|
| 0 | 0 | 0 | 0.010 |
| 0 | 0 | 1 | 0.008 |
| 0 | 1 | 0 | 0.006 |
| 0 | 1 | 1 | **0.005** |
| 1 | 0 | 0 | **0.005** |
| 1 | 0 | 1 | **0.005** |
| 1 | 1 | 0 | 0.006 |
| 1 | 1 | 1 | 0.008 |

## A.3 SELECTIVE USE OF ANTIFAULT

16-bit data types in AI models consumes less power during arithmetic operations compared to standard 32-bit data types. However, adding fault tolerance to these 16-bit numbers such as checking and correcting each bit can introduce some extra computational overhead. In the case of AntiFault, this overhead depends on how many parts (or modules) of the model are being protected using the AntiFault format.

AntiFault is flexible and does not need to be applied to every single parameter in the model. Instead, it can be used only on the most important or vulnerable layers of the AI model. This means that the less critical or more robust parts of the model can still use a simpler representation, such as a 16-bit format with 1 sign bit, 8 exponent bits, and 7 mantissa bits. These layers do not need the added fault protection, so they can operate more efficiently.

By selectively applying AntiFault only where it is truly needed, we can have more reliability with minimum computational overhead. This strategy helps minimize the overall computational overhead while still ensuring that the most sensitive parts of the model are protected from soft errors like single-bit flips.