# OpenReview forum: "AntiFault: A Fault-Tolerant and Self-Recoverable Floating-Point Format for Deep Neural Networks"
_ICLR.cc/2026/Conference — ICLR 2026 Conference Withdrawn Submission_

### Official Review · Reviewer_AM8b · 2025-10-31

**Soundness:** 2
**Presentation:** 4
**Contribution:** 1
**Rating:** 2
**Confidence:** 5

**Summary:**

The authors present a parity bit based number system, tailored to minimize the loss in resolution w.r.t. the regular operating range of neural network outputs and parameters, that claims fast, low-cost error detection and more importantly, exact restoration. Float32 numbers are converted to 16-bit resilient float numbers, passed through computations, and parity bits checked to ensure that computations within the neural network have not resulted in a single- or double-bit error. The authors have tested their system against limited bit errors in the inference phase of the neural network.

The paper suffers from a limited literature survey in the fault tolerance space (see below), and does not adequately compare against the large body of recent work in architecture and hardware that examines this very problem. It similarly does not assume multi-bit errors and does not provide a hardware-realistic fault model as opposed to random single or double bitflip injection.

**Strengths:**

The paper does provide a theoretically grounded parity bit based data format that should guarantee against single bit errors (prior art has examined correctability limits and this falls well within the information needed for single bit correction), and limited protection against double-bit flips (masking may occur within the parity bit resolution space where one bit flips one way and another flips the opposite). This is a recent space of concern, since many large distributed ML workloads are seeing one-in-a-billion bit errors more often, making this a very timely paper.

**Weaknesses:**

The paper has a few glaring issues, enumerated below:

1) Prior art:

-Algorithm-Based Fault Tolerance: The paper has cited Huang and Abraham (1987), but ignores work such as [1] that examines the number of bits necessary for correction of matrix computation errors - given that a limitation of the work is double bit errors and masking, [1] would allow the authors to clearly establish an analytical bound on their approach's effectiveness. Similarly, while the authors of ABFT papers have provided a thorough treatment of the overhead of their approach, the authors of this paper have not shown that their approach has no material effect on neural network latency.

-On-Line Fault Tolerance: [2] provides online updating statistical thresholds for error suppression in reinforcement learning systems that has been validated on hardware, [3] uses median feature selection to do the same thing, [4] makes use of statistics of successive differences, [5] simply uses a clipping threshold to set extreme values to zero. All of this work has not been mentioned or used as a baseline in this paper. All of these approaches have looked at the hardware overhead of their approach. I would ask that the authors please make use of either a baseline so that we can evaluate this tool in fairness against the state of the art, or that the authors contrast their tool against this state of the art qualitatively in the literature survey.

-Number Formats:  Resilient number formats for neural network training was examined in [6], error resilience of quantized number systems was examined in [7]. This is fundamentally a number formats paper and yet has not examined these baselines or mentioned them in the literature survey.

Please note that these are simply single papers. This alone, in a single review, should show that this is not an empty problem space. Which leads me to:

2) Baselines: The authors do not have a baseline in Table 1. There is simply a comparison to IEEE 754's float32 standard, which has been analyzed exhaustively in prior art for its accuracy issues as well as in the literature above for error resilience. That is not a fair baseline at all - especially when errors in higher exponent bits can simply be set to zero with little impact on classifier inference whether convolutional network, transformer or spiking network [8, 9, 10]. Without a baseline I have to state that this contribution is marginal because I have no evidence to the contrary.

3) Section 4.5 does not provide a summary of overhead numbers aside from a brief aside for memory overhead. Given that Anti fault seems to be generalizable to more - or less - than four parity bits, what is the Pareto tradeoff for detectability vs overhead? What is the theoretical overhead in operation count? What is the actual latency overhead for the given networks?

[1] Anfinson, Cynthia J., and Franklin T. Luk. "A linear algebraic model of algorithm-based fault tolerance." IEEE Transactions on Computers 37.12 (2002): 1599-1604.

[2] C. Amarnath and A. Chatterjee, "A Novel Approach to Error Resilience in Online Reinforcement Learning," 2023 IEEE 29th International Symposium on On-Line Testing and Robust System Design (IOLTS), Crete, Greece, 2023, pp. 1-7, doi: 10.1109/IOLTS59296.2023.10224892.

[3] Ozen, Elbruz, and Alex Orailoglu. "Boosting bit-error resilience of DNN accelerators through median feature selection." IEEE Transactions on Computer-Aided Design of Integrated Circuits and Systems 39.11 (2020): 3250-3262.

[4] C. Amarnath, M. Mejri, K. Ma and A. Chatterjee, "Soft Error Resilient Deep Learning Systems Using Neuron Gradient Statistics," 2022 IEEE 28th International Symposium on On-Line Testing and Robust System Design (IOLTS), Torino, Italy, 2022, pp. 1-7, doi: 10.1109/IOLTS56730.2022.9897815.

[5]Chen, Zitao, Guanpeng Li, and Karthik Pattabiraman. "A low-cost fault corrector for deep neural networks through range restriction." 2021 51st Annual IEEE/IFIP International Conference on Dependable Systems and Networks (DSN). IEEE, 2021.

[6] Long, Yun, Xueyuan She, and Saibal Mukhopadhyay. "Design of reliable DNN accelerator with un-reliable ReRAM." 2019 Design, Automation & Test in Europe Conference & Exhibition (DATE). IEEE, 2019.

[7] Domanski, Peter, et al. "Silent Data Corruption: Advancing Detection, Diagnosis, and Mitigation Strategies." 2025 IEEE 43rd VLSI Test Symposium (VTS). IEEE, 2025.

[8] Ozen, Elbruz, and Alex Orailoglu. "Just say zero: Containing critical bit-error propagation in deep neural networks with anomalous feature suppression." Proceedings of the 39th International Conference on Computer-Aided Design. 2020.

[9] K. Ma, C. Amarnath and A. Chatterjee, "Error Resilient Transformers: A Novel Soft Error Vulnerability Guided Approach to Error Checking and Suppression," 2023 IEEE European Test Symposium (ETS), Venezia, Italy, 2023, pp. 1-6, doi: 10.1109/ETS56758.2023.10174239.

[10] A. Saha, C. Amarnath and A. Chatterjee, "A Resilience Framework for Synapse Weight Errors and Firing Threshold Perturbations in RRAM Spiking Neural Networks," 2023 IEEE European Test Symposium (ETS), Venezia, Italy, 2023, pp. 1-4, doi: 10.1109/ETS56758.2023.10174229.

**Questions:**

1) Would the authors be able to contrast their work against the survey above? At the very least, a qualitative correction is needed - I understand that additional data is not a feasible request in this conference, given response times.

2) Would the authors be able to provide a baseline? Even ABFT in Table 1 as a baseline would be good, but simply comparing against 754 is absolutely not a fair comparison or test of the system.

3) What was the fault rate and model? Was this mostly focused on particle strike errors, which cause 1-to-0 flips due to capacitor discharge? Thermal or wear induced errors, timing errors, which are more uniform? A discussion of that is absolutely needed for a SDC paper.

4) Inference time errors are often not in full precision float32 for large, modern language models since language models are often quantized for cost and speed reasons. Do the reviewers have a comment as to how their system would generalize to float16, BFloat16, float8, int8, int4 or mxp?

---

### Official Review · Reviewer_asvH · 2025-11-01

**Soundness:** 2
**Presentation:** 1
**Contribution:** 1
**Rating:** 0
**Confidence:** 4

**Summary:**

The paper tackles the soft error in the memory system, with a specific focus on AI model inference. The authors address this issue by proposing a novel floating-point representation that essentially i) compresses the exponent field by building a look-up table, ii) compresses the mantissa field via truncation of LSBs, and iii) adds a parity bit for error detection and correction.

**Strengths:**

The paper seems to provide comprehensive experimental results, although limited to small models and datasets, not even including ImageNet.

**Weaknesses:**

- I am not convinced with the motivation of this work. Our memories seem to be already very reliable with little to no error. Under what context are these soft errors prevalent, leading to model collapse, hence making this research relevant and impactful?
- As an extension of the above point, are the BERs assumed in the experimental section realistic? DRAMs of CPUs and VRAMs of GPUs seem to exhibit much lower BERs (1 bit error per gigabyte of RAM per 1.8 hours). This is also true heuristically; I have never seen an image classification model experience such severe accuracy degradation in real life. The authors should better justify this assumption and its validity.
- Also, modern server-class GPUs, such as A100s and H100s, are equipped with ECC memory, which is already capable of error correction. This also seems to undermine the necessity of the proposed number format.
- The authors claim the proposed number format reduces model size by 50% and pose it as their contribution. I am uncomfortable with this argument, as the model tested in the papers are highly redundant and overparametrized models that can be easily quantized to ultra-low precision. Hence, it seems unfair to claim 50% compression comparing with the FP32 model; the authors should instead compare their work in a more realistic setting, FP8 or INT8 quantized model.
- I am not sure if the AI conference is the right place for this paper, as it is not exactly pushing the frontiers of AI research, but rather in the field of computer architecture and systems, with an application of AI. The research work will be better appreciated by that community.

**Questions:**

No questions

---

### Official Review · Reviewer_sjR5 · 2025-11-01

**Soundness:** 3
**Presentation:** 3
**Contribution:** 3
**Rating:** 4
**Confidence:** 5

**Summary:**

This paper introduces AntiFault, a new 16-bit number format that is influenced by the floating point standard but with fault-tolerance at the core. The new number format includes small approximations to the orignal value, and reserves multiple bits for fault tolerance (Hamming Code / Parity). The results are decent, but mostly on older models/networks.

**Strengths:**

+ Fault tolerance is very important, in particular for today's datacenters.
+ Thinking about fault tolerance at the number format level is very fundamental and good.
+ The number format itself is a bit unique, and I like it. Most innovations at this level (such as posit's or mxfp) really focus on performance; however this idea focuses on maintaining performance while providing robustness.

**Weaknesses:**

- There are a lot of new number formats coming out these days actually, but none are acknowledged or discussed here. This includes things such as Block Float derivatives (Mxfp), which is now an OCP standard; AdaptivFloat, which is also adopted by tesla; and various other formats such as FP8, which are FP-derivatives. As is, AntiFault is evaluated and proposed in isolation, and hard to understand the relevance to other related work empirically as well as qualitatively. Despite other formats being performance-centric, recent tools have begun evaluating their robustness to errors (such as GoldenEye [DSN '22]). Why would someone adopt a new hardware numerical type (AntiFault), if they can work around the weaknesses of existing number formats (MXPF, FP8, etc)?
- Another weakness is the generalizability of this format is not discussed. Floating Point, to it's credit, allows a designer to modify the exp to mantissa ratio, and consequently change the precision/range, and also the bitwidth. This is how many folks have come up with variable number formats (as FP derivatives). However, AntiFault seems to be presented in only a 16bit format, without much discussion on the implications of increasing/decreasing bitwidth, or other smaller changes (how many bits to allocate for exp, for example), and the impact on fault tolerance. This is a critical missing point, and really should be incorporated for any number format discussion.
- No comparison with modern LLMs; most networks are a bit older. This is not a dealbreaker, but it would be much nicer if more recent LLMs were included.

**Questions:**

- Comparison to other number formats. Why would someone choose AntiFault, rather than FP8 (for example) and duplicate FP8 (so still 16 bits wide)? This is a bit simple to the reviewers mind, but can the authors address it in terms of more recent number formats like MXFP?
- How does this number format "adapt", or is it stuck at 16 bits?

---

### Official Review · Reviewer_vHT2 · 2025-11-01

**Soundness:** 3
**Presentation:** 3
**Contribution:** 2
**Rating:** 2
**Confidence:** 4

**Summary:**

The paper presents AntiFault, a novel 16-bit floating-point representation designed to provide fault tolerance and self-recovery for Deep Neural Networks (DNNs) against soft errors. AntiFault works by approximating 32-bit floating-point numbers into a 16-bit format and embedding fault detection and correction mechanisms, using parity bits derived from the Hamming code, within this same 16-bit format. The results for ResNet18, MobileNetV2, MobileViT, DistilBERT, and RoBERTa models demonstrate that the proposed representation can help mitigate soft errors at moderate fault rates.

**Strengths:**

1. The paper presents a novel 16-bit floating-point representation designed to provide fault tolerance and self-recovery for Deep Neural Networks (DNNs) against soft errors.
2. The paper addresses a core issue in the reliable inference of DNNs, namely the problem of data corruption due to soft errors.
3. The paper presents results for a diverse set of models, highlighting that the proposed method can mitigate soft errors while generating reliable results at moderate fault rates.

**Weaknesses:**

1. It is unclear whether the paper specifically targets soft errors in the weight memory of hardware accelerators or elsewhere. Additionally, there is no discussion about why only a single memory component requires protection, and not others.
2. The motivation behind focusing on the 16-bit floating-point format is not clear. The paper focuses on mitigating soft errors during DNN inference; however, the data format targeted is the 16-bit floating-point format. Also, due to the reduction from 32 bits to 16 bits for each weight, the paper claims memory savings, which seems strange, as inference can be performed using 8-bit precision as well without any significant reduction in model accuracy. In such a case, even a version of the DMR approach would result in reasonable mitigation with minimal accuracy loss.
3. Literature coverage is very limited. DMR and TMR are very old techniques and there is a lot of work in the literature on low-cost soft-error mitigation for reliable DNN inference.
4. Comparison with the state-of-the-art is minimal. The authors should have considered other techniques as well, such as Chen, Zitao, Guanpeng Li, and Karthik Pattabiraman. "A low-cost fault corrector for deep neural networks through range restriction." 2021 51st Annual IEEE/IFIP International Conference on Dependable Systems and Networks (DSN). IEEE, 2021, to demonstrate the significance of the overheads for mitigating soft errors. The authors should explain that the proposed technique offers the best cost-to-mitigation trade-off.

**Questions:**

1. Were all bit positions subjected to bit-flip errors for simulation experiments?
2. How does the proposed technique compare with other state-of-the-art techniques, such as Chen, Zitao, Guanpeng Li, and Karthik Pattabiraman "A low-cost fault corrector for deep neural networks through range restriction." 2021 51st Annual IEEE/IFIP International Conference on Dependable Systems and Networks (DSN). IEEE, 2021? How can the authors justify the overhead cost compared to the state-of-the-art methods, specifically in terms of energy and latency?
3. Is it assumed that soft errors can only occur in the weight memory of the hardware? If so, can the proposed technique be applied in any form to protect other hardware modules, such as computational units and activation buffers? If no, then scope of the proposed technique seems very limited.
4. How can the authors justify the use of 16-bit floating-point format when the inference can be performed using 8-bit or less precision as well? Even the increased bit precision should be considered a part of the overhead cost.

---

### Note · Authors · 2025-11-18

I have read and agree with the venue's withdrawal policy on behalf of myself and my co-authors.